# Rewritable ghost floating gates by tunnelling triboelectrification for two-dimensional electronics

Seongsu Kim[1,*], Tae Yun Kim[2,*], Kang Hyuck Lee[1], Tae-Ho Kim[1], Francesco Arturo Cimini[3], Sung Kyun Kim[1], Ronan Hinchet[1], Sang-Woo Kim[1,2] & Christian Falconi[3]

Gates can electrostatically control charges inside two-dimensional materials. However, integrating independent gates typically requires depositing and patterning suitable insulators and conductors. Moreover, after manufacturing, gates are unchangeable. Here we introduce tunnelling triboelectrification for localizing electric charges in very close proximity of two-dimensional materials. As representative materials, we use chemical vapour deposition graphene deposited on a $SiO_2$/Si substrate. The triboelectric charges, generated by friction with a Pt-coated atomic force microscope tip and injected through defects, are trapped at the air–$SiO_2$ interface underneath graphene and act as ghost floating gates. Tunnelling triboelectrification uniquely permits to create, modify and destroy p and n regions at will with the spatial resolution of atomic force microscopes. As a proof of concept, we draw rewritable $p/n^+$ and $p/p^+$ junctions with resolutions as small as 200 nm. Our results open the way to time-variant two-dimensional electronics where conductors, p and n regions can be defined on demand.

[1] School of Advanced Materials Science and Engineering, Sungkyunkwan University (SKKU), 2066 Seobu-ro, Jangan-gu, Suwon, Gyeonggi-do 440-746, Republic of Korea. [2] SKKU Advanced Institute of Nanotechnology, Sungkyunkwan University (SKKU), Suwon 440-746, Republic of Korea. [3] Department of Electronic Engineering, University of Rome Tor Vergata, Via del Politecnico 1, Roma 00133, Italy. * These authors contributed equally to this work. Correspondence and requests for materials should be addressed to S.-W.K. (email: kimsw1@skku.edu) or to C.F. (email: falconi@eln.uniroma2.it).

In conventional field-effect devices[1,2], the gate voltage electrostatically controls charge carriers inside a semiconductive channel, which is separated from the gate by an insulating layer. This gate-driven electrostatic control of charge carriers has been demonstrated for graphene[3,4] and other two-dimensional (2D) materials[5,6]. Moreover, it is relatively easy to integrate on a single chip many 2D field-effect devices sharing a single global bottom gate. However, the very large-scale integration (VLSI) of independent field-effect devices on a single chip is very complex. In fact, similar to state-of-the-art complementary metal-oxide semiconductor (CMOS) devices, the fabrication of independent gates, insulating layers and channels requires the growth and patterning of different, high-quality and carefully selected materials[7,8], with intricate process-compatibility issues. This problem is exacerbated by severe geometrical constraints; in fact, besides the quest for ultra-small gate dimensions, the insulating layer must be extremely thin to guarantee an effective (that is, with reasonably low voltages) electrostatic induction of charges in the channel. In addition, in conventional electronics, once the circuit has been manufactured, its structure and, in particular, the positions and shapes of all the gates may not be modified.

Triboelectrification is the electrical charging by friction between two materials[9–11]. Although the triboelectric effect is reported to have been first observed by Thales of Miletus, it is still a subject of intense research and, recently, has been investigated, with nanoscale spatial resolution, by using atomic force microscopy (AFM). In practice, by rubbing insulators with the tips of AFM[12,13], electrical charges can be localized on insulators and be stored for relatively long periods, around 1 h.

Here we introduce tunnelling triboelectrification for defining on-demand rewritable ghost floating gates below a 2D material with the spatial resolution of AFMs[14,15]. Tunnelling triboelectrification is the friction-induced tunnelling of charges through a 2D material and their accurate localization on the insulator underneath the 2D material. Tunnelling of charges may also occur in conventional triboelectrification processes, but in tunnelling triboelectrification charges tunnel through a 2D material rather than simply through air or vacuum. Moreover, though charges can be localized even by conventional triboelectrification of dielectrics such as $SiO_2$ (refs 12,13), the charges injected by tunnelling triboelectrification exhibit impressively longer lifetimes (for example, more than two order of magnitude longer). Finally, after tunnelling triboelectrification, the charges very effectively control the properties of the 2D material, thus behaving as immaterial, charges-only, ghost floating gates, which can be repeatedly created, modified or destroyed; this unique property may be the key for the development of novel 2D devices, which can be drawn or modified on demand.

## Results

### Tunnelling triboelectrification.
We deposited CVD graphene on a $SiO_2$ (300 nm)/Si wafer using a wet transfer method. As schematically shown in Fig. 1a, with silicon connected to ground, we rubbed the CVD graphene over a $0.5 \times 0.5\,\mu m^2$ area using a grounded Pt-coated AFM tip in contact mode, with a force of 15 nN. We verified by AFM that rubbing does not result in detectable mechanical damages to graphene (Supplementary Note 1 and Supplementary Fig. 1) and also measured the thickness of the unavoidable air-gap between graphene and $SiO_2$ substrate, which is around 0.66 nm[16,17] (Supplementary Fig. 2). To measure the surface potential, both before and after triboelectrification[18,19], we have used Kelvin probe force microscopy (KPFM)[20,21]. Figure 1b shows the initial state of the CVD graphene surface potential on a $1.5 \times 1.5\,\mu m^2$ area; as evident, before rubbing, graphene is equipotential, except random fluctuations (for example, due to undesired trapped charges and contamination). Figure 1c shows the surface

potential in the entire $1.5 \times 1.5\,\mu m^2$ area after rubbing the $0.5 \times 0.5\,\mu m^2$ central square; the rubbed region has a $\sim 50\,mV$ higher surface potential than the unrubbed region. Such potential variation may not be attributed to charges stored in the graphene as electric charges may be localized for long times only in insulating materials. We therefore conclude that some of the charges generated during rubbing tunnel through the monolayer graphene and are locally trapped on the underlying insulator. The trapped charges act as an immaterial (that is, made of charges-only and not of a conductor), bottom floating gate and locally change the polarity and density of charges in graphene as well as the graphene work-function; for simplicity we will therefore refer to these trapped charges as to ghost floating gates. The ghost floating gates effectively control the carriers within graphene because of the ultimate thinness of graphene as well as of the comparably thin air-gap.

Figure 1d shows the averaged $\Delta V_{TT}$ (across the central $0.5\,\mu m$ horizontal stripe; the average is computed to reduce the effect of random fluctuations) as a function of position (along the black dashed line of Fig. 1c) taken immediately after rubbing and after 72 h (Supplementary Fig. 4 shows the original KPFM maps), where, to highlight the effects of rubbing rather than random fluctuations, $\Delta V_{TT}$ is defined as the surface potential taken with reference to the average surface potential of the unrubbed region (that is, $\Delta V_{TT}$ shows the net variations of the surface potential, due to tunnelling triboelectrification, in the rubbed region in comparison with the unrubbed region). As evident, the potential variation is very well preserved even after 72 h. Figure 1e shows the averaged $\Delta V_{TT}$ as a function of time and the best fit function (blue line)

$$\Delta V_{TT}(t) \simeq 4mV * e^{\frac{-t}{\tau_{short}}} + 46mV * e^{\frac{-t}{\tau_{long}}}. \tag{1}$$

In practice, along with a small term, which has a shorter time constant ($\tau_{short} \sim 3\,h$ and 21 min), there is a dominant term with much higher initial amplitude and with an exceptionally long time constant ($\tau_{long} \sim 278\,h$, that is, more than two orders of magnitude higher than the decay time, around $1\,h^{13}$, of conventional triboelectrification on a $SiO_2$ dielectric with the same thickness, 300 nm, as in our experiments, Supplementary Note 3.3 and Supplementary Fig. 10). The Supplementary Note 2 and Supplementary Figs 3–6, 13 show additional experiments which further confirm the proposed tunnelling triboelectrification mechanism.

To gain further insight on tunnelling triboelectrification, we also studied the cases of monolayer (1L), bilayer (2L), trilayer (3L) CVD graphene and highly ordered pyrolytic graphite (HOPG). Similar to the experiments described in Fig. 1, we used $SiO_2$ (300 nm)/Si substrates, and measured the surface potential of a $2 \times 2\,\mu m^2$ graphene layer both before and after rubbing the $1 \times 1\,\mu m^2$ central area with a Pt AFM tip. Figure 2a shows, for each sample, the average surface potential changes which result from rubbing ($\Delta V_{after-before}$), both in the rubbed ($\Delta V_r$) and in the unrubbed ($\Delta V_u$) parts of the graphene films. $\Delta V_r$ (red line in Fig. 2a) decreases as the number of graphene layers increases; by contrast, $\Delta V_u$ (black line in Fig. 2a) increases and, for highly ordered pyrolytic graphite (HOPG), both $\Delta V_r$ and $\Delta V_u$ converge to the same value, similar to what would happen without tunnelling triboelectrification, that is, the film would be equipotential (except random fluctuations) and the triboelectric charges would simply charge the graphene–air–$SiO_2$–silicon capacitor. Except for HOPG, the graphene layers are not equipotential as the trapped charges, similar to the presence of ghost floating gates at the $SiO_2$–air interface, locally increase or decrease, depending on their polarities, the graphene potential. In practice, tunnelling triboelectrification and the spread of charges on the entire graphene layer are two competing processes.

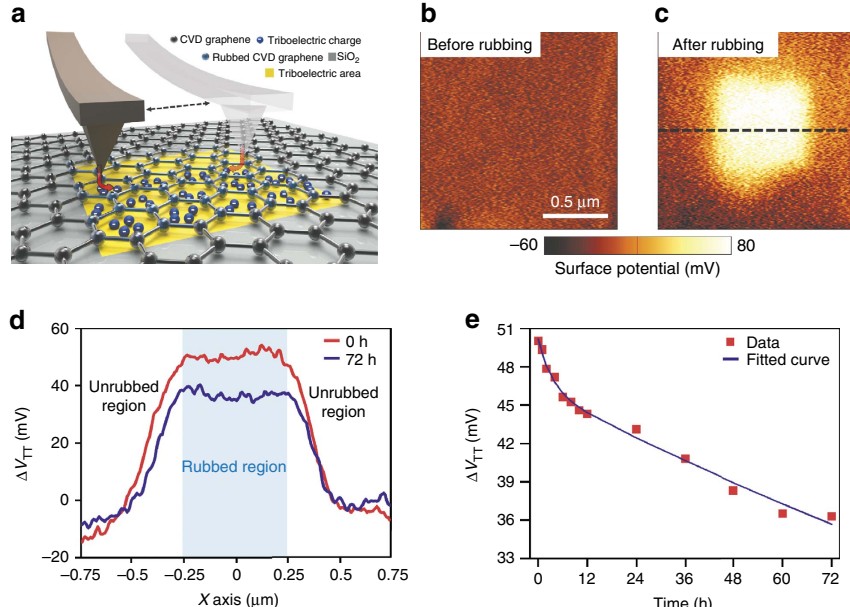

**Figure 1 | Tunnelling triboelectrification by friction of graphene with a Pt AFM tip.** (**a**) Schematic of the friction process and the KPFM measurement system. (**b**) KPFM image of graphene before rubbing. (**c**) KPFM image of graphene after rubbing. (**d**) Potential difference generated by tunnelling triboelectrification, $\Delta V_{TT}$, along the blue dashed line in **c** after 0 and 72 h; $\Delta V_{TT}$ is very well preserved even after 72 h. (**e**) $\Delta V_{TT}$ as a function of time and best fit (blue line) with the sum of two decaying exponential terms, each with its own time constant.

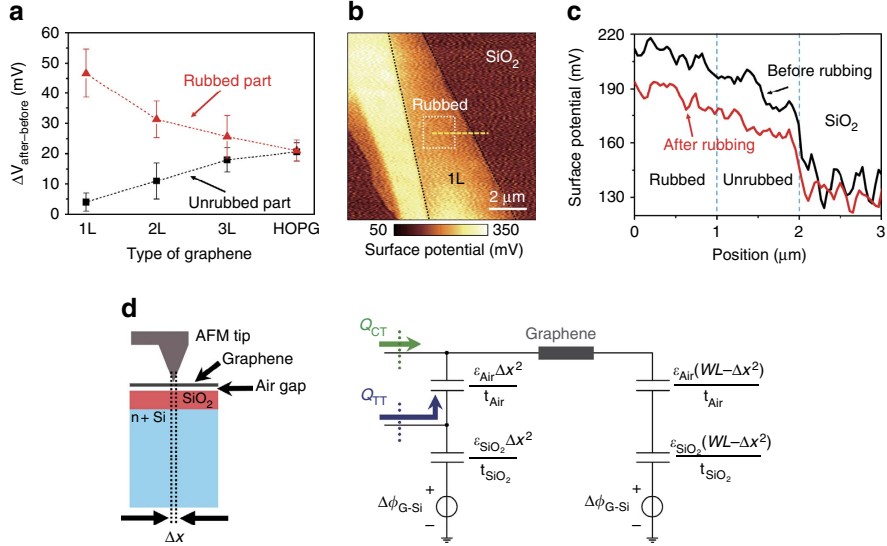

**Figure 2 | Tunnelling triboelectrification with different types of graphene and equivalent circuit.** (**a**) $\Delta V_r$ and $\Delta V_u$ for different types of CVD graphene (1, 2 or 3 layers) and for HOPG. (**b**) KPFM image of mechanically exfoliated graphene (MEG) after rubbing with an AFM tip. (**c**) KPFM image of the MEG layer measured along the yellow dashed line in **b** and showing that there is no significant potential difference between the rubbed and unrubbed parts (the almost constant shift is due to triboelectric charges spreading all over the graphene layer and thus charging the graphene-to-silicon capacitor). (**d**) Schematic diagram (not to scale) and simplified, lumped elements equivalent circuit for the tunnelling triboelectrification process; the equivalent circuit comprises the small areas ($\Delta x^2$) air-gap and $SiO_2$ capacitors (rubbed section) and the large area capacitors (rest of the graphene layer). Error bars are defined as s.e.m.

The spread of charges involves the entire graphene layer and is global; on the contrary, tunnelling triboelectrification is restricted to an area of graphene comparable with the contact area of the AFM tip and, therefore, is extremely localized. As evident from Fig. 2a, for 1L CVD graphene the tunnelling triboelectrification charges are almost identical to the entire triboelectric charges; in fact, for monolayer CVD graphene, tunnelling triboelectrification prevails over the spread of charges on the entire graphene layer because of the ultimate thinness of graphene as well as of the

extremely high speed of tunnelling processes, which take much less time than charging the graphene–air–$SiO_2$–silicon capacitor. By contrast, when increasing the number of layers, tunnelling becomes more and more unlikely and triboelectric charges tend to spread over the entire graphene layer and, consequently, increase or decrease the potential of the entire graphene layer.

We also performed similar experiments on mechanically exfoliated graphene on $SiO_2$/Si substrates. Figure 2b shows the surface potential of monolayer exfoliated graphene (MEG)

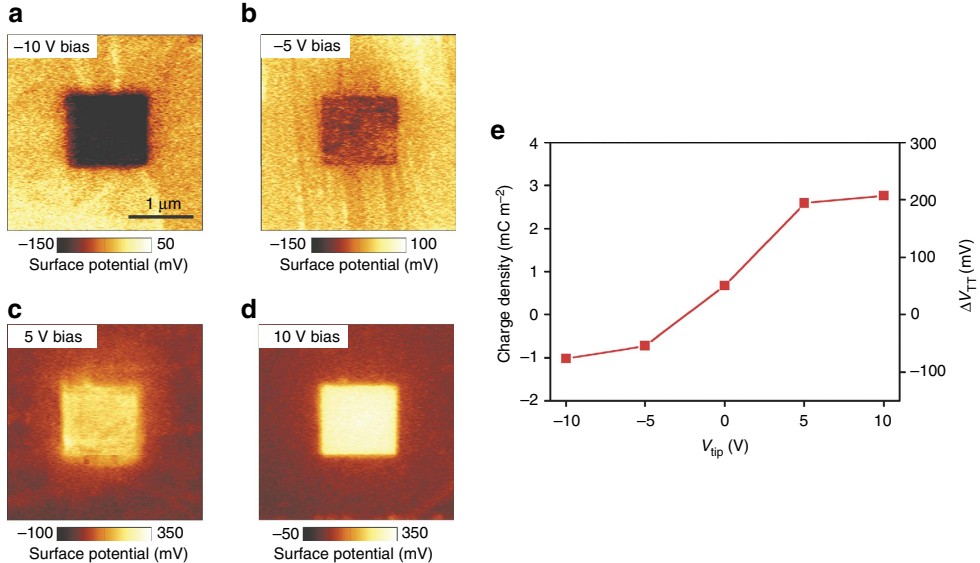

**Figure 3 | Control of the density and polarity of the tunnelling triboelectric charges. (a–d)** KPFM images of graphene after rubbing the central area with different tip bias voltages of (**a**) − 10 V, (**b**) − 5 V, (**c**) 5 V, (**d**) 10 V. (**e**) Triboelectric charge density and potential drop induced by tunnelling triboelectrification ($\Delta V_{TT}$) as a function of the tip bias. The polarity and amount of the tunnelling triboelectric charges can be controlled by the tip bias voltage.

(black dashed line region) after rubbing an area of $1.5 \times 1.5\,\mu m^2$ (white box region) with the Pt AFM tip. As shown in Fig. 2c, contrary to CVD graphene, triboelectric charges do not tunnel and, except random fluctuations, there is no detectable potential difference between the rubbed and unrubbed areas. Instead, the surface potential of the entire MEG region, both rubbed and unrubbed parts, changes by a constant quantity, similar to the case of HOPG. These results further confirm the proposed mechanism because defect regions such as grain boundaries have higher empty state tunnelling transmission coefficients than bulk graphene[22,23], so that charges generated by rubbing can more easily pass through the defective 1L CVD graphene. By contrast, MEG, though monolayer, has less defects (Supplementary Fig. 7) and, therefore, the triboelectric charges diffuse along the whole MEG layer. In conclusion, the triboelectric charge localization is most effective in 1L CVD graphene due to both its single-atomic thickness and high-defect density.

Figure 2d schematically illustrates that the tunnelling triboelectrification process is confined in an extremely small area with a characteristic length $\Delta x$ comparable with the very sharp tip of the AFM. In the simplified, lumped elements, equivalent circuit for tunnelling triboelectrification shown in Fig. 2d, we distinguish the small area, rubbed section (capacitors with area $\Delta x^2$) and the global section (capacitors relative to the rest of the graphene layer, with area approximately equal to the entire graphene area, $WL$). Owing to rubbing, a fraction of the triboelectric charges tunnel through graphene and is injected at the interface between air and $SiO_2$ (tunnelling triboelectric charges, $Q_{TT}$); the rest of the triboelectric charges (complementary-triboelectrification charges, $Q_{CT}$) spreads all over the graphene layer. Immediately after tunnelling, the charges $Q_{TT}$, trapped at the air–$SiO_2$ interface, electrostatically attract charges of the opposite type on the graphene layer and/or on the silicon underneath silicon oxide (that is, the charges $Q_{TT}$ may not travel through dielectrics and therefore must be stored on the top plate of the $SiO_2$ small-area capacitor and/or on the bottom plate of the small-area air capacitor). However, as graphically illustrated in Fig. 2d, almost all these opposite charges are attracted from graphene because of the much smaller thickness (that is, larger capacitance) of the small-area air capacitor in comparison with the small-area $SiO_2$

capacitor (Supplementary Note 3 and Supplementary Figs 8–10). With reference to the equivalent circuit, the charges $Q_{TT}$ are almost all stored on the bottom plate of the small-area air capacitor and then act as a ghost floating gate which is not manufactured but can be triboelectrically drawn and deleted at will by the AFM tip. Since almost no charges go towards the small-area $SiO_2$ capacitor, its voltage is almost unaffected and, therefore, the voltage drop across the small-area air capacitor is almost exactly the same as the voltage drop measured across graphene by KPFM, that is, $\Delta V_{TT}$. Remarkably, since the air-gap is extremely thin (about 0.66 nm, that is, comparable with the equivalent oxide thicknesses of state of the art CMOS devices, Supplementary Fig. 2), the charges $Q_{TT}$ very effectively control the current transport inside graphene. Moreover, despite the very small air-gap thickness, the ghost floating gate does not introduce significant parasitic capacitance as, dynamically, the series of the oxide and air capacitance is almost identical to the silicon oxide capacitance alone. The equivalent circuit shown in Fig. 2d also gives reasons for the exceptionally long decay time of tunnelling triboelectrification charges (Fig. 1e). In fact, since, after triboelectrification, there is no current flow through graphene (the potential drop across graphene may not result in a net current flow because of the competing electrostatic attraction from the charges stored on the immaterial ghost gate), both the air-gap and the $SiO_2$ capacitors can only be discharged by their leakage currents (Supplementary Note 3.3), thus resulting in two time constants associated to the air-gap and the $SiO_2$ capacitors, respectively. In conclusion, first, in regions subject to tunnelling triboelectrification, the air-gap capacitors are charged to comparatively much higher voltages than $SiO_2$ capacitors because the tunnelling triboelectric charges almost entirely go towards the bottom plates of air-gap capacitors. Second, the total voltage is the sum of two terms, each with its own time constant; the dominant term, relative to the air-gap capacitor, has a much larger time constant because air is a much better insulator than $SiO_2$, in perfect agreement with our experimental results (two time constants best fit in Fig. 1e). The excellent insulation properties of air also justify the exceptionally long decay of the tunnelling triboelectrification voltages in comparison with conventional triboelectrification[12,13].

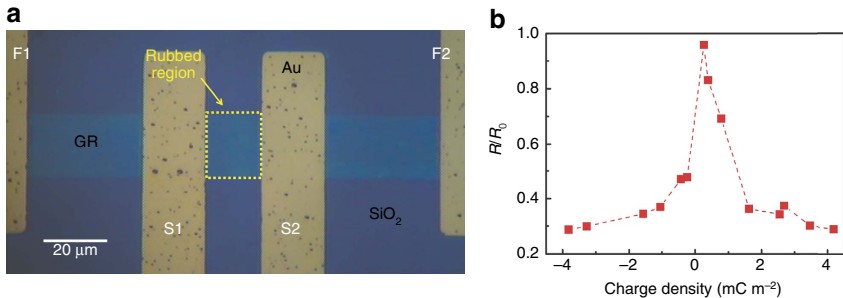

**Figure 4 | Resistance reduction of graphene by tunnelling triboelectrification.** (**a**) Optical image of the 4-wires graphene resistor (the yellow dashed box is rubbed by the Pt AFM tip). (**b**) Normalized resistance $R/R_0$ of the graphene resistor as a function of the tunnelling triboelectric charge density.

In conventional triboelectrification, both the density and polarity (positive or negative) of the triboelectric charges injected into a rubbed insulating material can be controlled by the application of a biasing voltage $V_{tip}$ to the Pt AFM tip during rubbing[13]. Therefore, since the tunnelling triboelectrification charges $Q_{TT}$ are a fraction of the total triboelectric charges, both the density and polarity of the trapped charges as well as the sign of the voltage drop between the rubbed and the unrubbed parts of graphene can be controlled by changing the tip biasing voltage. Consistently, Fig. 3a–d show $\Delta V_{TT}$ after the central square has been rubbed with the Pt AFM tip biased at voltages ranging from $-10$ to $+10$ V. At negative biasing voltages (Fig. 3a,b), negative charges are trapped and attract holes in the rubbed part of the graphene, thus increasing the natural p-type conductivity of CVD graphene. On the contrary, at positive biasing voltages (Fig. 3c,d), positive charges are injected and attract electrons in the rubbed part of the graphene so that, at sufficiently high biasing voltages, graphene is inverted to n-type conductivity. Figure 3e shows the average surface potential of the rubbed part, taken with reference to the unrubbed part, as a function of $V_{tip}$. Since the surface potential, taken with reference to the potential of the unrubbed part, is almost identical to the voltage locally stored across the small-area air-gap capacitors, the tunnelling triboelectric charges surface density (Fig. 3e) can be computed as the voltage drop across the air-gap capacitor multiplied by the air-gap capacitance per unit area $\varepsilon_{Air}/t_{Air}$, where $\varepsilon_{Air}$ is the dielectric constant of air and $t_{Air}$ is the thickness of the air-gap.

Tunnelling triboelectrification also allows to control the resistivity of graphene. As an example, we fabricated the 4-contacts 1L CVD graphene resistor shown in Fig. 4a. The middle area of the graphene resistor (yellow dashed box, $17 \times 20\,\mu m^2$, between the sense electrodes S1 and S2 in Fig. 4a) was rubbed by a Pt tip. The voltage difference of the 1L CVD graphene between the inner electrodes S1 and S2 was measured in vacuum by applying a constant current through the external force electrodes F1 and F2. During the measurements, we kept the current at a very low level (that is, 100 nA) to prevent any perturbation of the stored charges. In fact, the tunnelling triboelectric charges stored underneath graphene may be removed at sufficiently high currents. Although the threshold current depends on the sample, we found that trapped charges may survive even under currents up to the mA range for widths of only 20 $\mu$m (Supplementary Fig. 11). Figure 4b shows the normalized resistance $R/R_0$ (where $R$ is the resistance of graphene after rubbing and $R_0$ is the resistance before rubbing) as a function of the tunnelling triboelectric charge density (See Supplementary Note 4.1 and Supplementary Fig. 12 for details). As expected, the tunnelling triboelectric charges act as ghost floating gates and electrostatically reduce the resistivity (for example, up to three times at room temperature with magnitude of the biasing tip voltage limited to 10 V). The dependence of

the resistance on the tip voltage is obviously similar to the gate bias-dependent resistance in graphene field-effect devices[3,24] with conventional gates. Moreover, we also verified that tunnelling triboelectrification allows to control the Dirac point of graphene in practical graphene field effect devices (Supplementary Note 4.2 and Supplementary Fig. 13).

**Rewritable floating gates**. Tunnelling triboelectrification allows the local and dynamic control of both the polarity and the density of free carriers in 2D materials with the extraordinary spatial resolution of AFMs. In particular, the dimensions and shapes of ghost floating gates or, equivalently, of p and n regions as well as of conductors (for example, made of very highly doped regions) can be dynamically changed over time, possibly resulting in truly time-variant electronic devices and systems. Since AFMs can be fully integrated on a single chip[25–27], we envision single chips comprising the AFM and regions of 2D materials whose properties can be controlled on demand by tunnelling triboelectrification. This approach would be similar to the memory refresh process (that is, periodically reading and imme- diately rewriting the same information on capacitive memories to counteract the degradation of information due to leakage currents) routinely used in dynamic random-access memories (DRAM). We also observe that tunnelling triboelectrification would also allow to simultaneously take advantage of different 2D materials on a single substrate without the intricacies of co-integrating suitable gates and insulating layers for different 2D materials (to manufacture conventional gates, each 2D material would require its own processing, including patterning of carefully selected materials as both insulators and gates). As a proof of concept, Fig. 5 shows several time-variant p/p$^+$ (Fig. 5a) and p/n$^+$ (Fig. 5b) junctions with resolutions as small as 200 nm. These p/p$^+$ and p/n$^+$ junctions are associated to rewritable ghost floating gates that can be repeatedly created, modified and erased. This is the first report of gates, p/p$^+$ and p/n$^+$ junctions whose shapes and charges can be defined on demand with deep sub-micron resolution.

**Discussion**
In conclusion, we have introduced tunnelling triboelectrification for dynamically localizing charges on immaterial, floating, bottom ghost gates underneath 2D materials. We rubbed graphene with a Pt-coated AFM tip and found that a fraction of the triboelectric charges, generated by friction between graphene and the AFM tip, tunnel through graphene and are trapped at the interface between the air-gap and the underlying insulator (SiO$_2$, mica or Al$_2$O$_3$ in our experiments). Tunnelling triboelectrification occurs in defective CVD graphene, especially for single-layer samples, but not in high-quality exfoliated graphene or in HOPG. We also

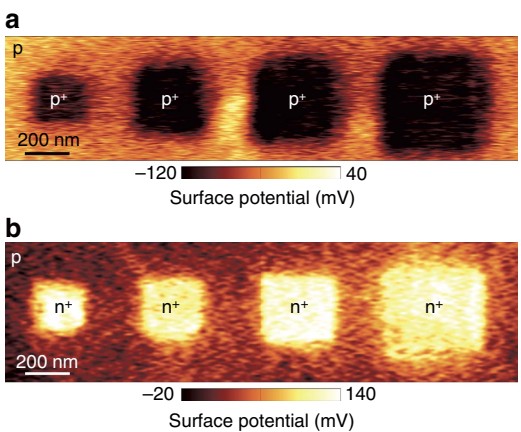

**Figure 5 | Rewritable p/p$^+$ and p/n$^+$ junctions and ghost floating gates.** (**a**) KPFM image of four deep sub-micron p/p$^+$ junctions defined by selectively rubbing arbitrary square graphene regions with the AFM tip biased by −10 V; each p/p$^+$ junction corresponds to an underlying ghost rewritable floating gate. The sides of each rubbed region are 200, 300, 400 and 500 nm, respectively, from left to right. (**b**) KPFM image of four deep sub-micron p/n$^+$ junctions defined by selectively rubbing arbitrary square graphene regions with the AFM tip biased by 10 V. The sides of each rubbed region are 200, 300, 400 and 500 nm, respectively, from left to right.

found a decay time of several days, which is more than two orders of magnitude longer than for standard triboelectrification. Owing to the ultimate thinness of 2D materials and to the sub-nanometer insulating air-gap, the immaterial ghost gates very effectively control charges inside graphene. In striking contrast with conventional electronic devices which, after manufacturing, may not be changed, the ghost floating gates as well as the resulting p and n regions can be created, enlarged, reduced or destroyed on-demand, with the resolution of AFMs and without the intricate process-compatibility issues for integrating many independent conventional field-effect devices on a single chip. Although these immaterial ghost gates are floating, with obvious restrictions for the design of analogue and digital circuits[28–32], other types of devices and transistors (for example, bipolar junction transistors) can also be fabricated by taking advantage of ghost floating gates. We also mention that tunnelling triboelectrification may easily be combined with conventional microfabrication technologies; for instance, in our experiments, we have deposited graphene on a conventional SiO$_2$/Si substrate, which is obviously compatible with the integration of CMOS analogue/digital circuits and/or MEMS, including micromachined AFMs. In fact, since AFMs can be fully integrated on a single chip[25–27], in contrast with conventional 'time invariant' micro devices whose gates, p and n regions are unchangeable, we envision systems-on-chip[33–39] or systems-in-package[40], where ghost floating gates and n and p regions are continuously, on-demand defined, modified or destroyed by tunnelling triboelectrification. As a proof of concept, we have dynamically defined several p/n$^+$ and p/p$^+$ junctions with resolutions as small as 200 nm. Our results may greatly facilitate the very large-scale integration of independent 2D field-effect devices and can open the way to time-variant 2D electronics, where conductors (for example, very highly doped regions), p and n regions can be defined on demand.

## Methods

**Synthesis of CVD graphene and transfer method.** A 75 μm-thick copper foil (Wacopa) was used for graphene growth. The copper foil was placed in a rapid thermal chemical vapour deposition (CVD) chamber, where the temperature was increased from room temperature to 1,000 °C under a 10 sccm flow of H$_2$ (1 Torr). Then, the copper foil was annealed for 30 min to clean its surface. The CVD graphene was synthesized by a mixture of CH$_4$ (20 sccm) and H$_2$ (10 sccm) for 30 min (1 Torr). After the growth was complete, the gas supply was ceased and the chamber was cooled below 100 °C at a cooling rate of 160 °C min$^{-1}$. The CVD graphene synthesized on copper foil was spin-coated with poly(methyl methacrylate) (PMMA) using spin coater with 1,000 r.p.m., 30 s and it was cured at 120 °C for 10 min. Then, the CVD graphene on copper foil was floated in an etchant (Transene, type 1) to etch away the copper foil. After the copper foil was completely etched away, the graphene with PMMA was rinsed in deionized water three times to wash away the etchant residues. The CVD graphene was transferred onto SiO$_2$ (300 nm)/Si (boron doped p-type, resistivity is 70 Ohm cm) substrate using a well-known wet transfer method. Multi-layer CVD graphene was prepared by repeating the same process[11].

**Generation and measurement of triboelectric charges.** The rubbing process between a Pt-coated tip (Multi75E-G, Budget Sensors) and graphene was carried out by the contact mode of an AFM system (XE100, Park Systems) under a contact force of 15 nN and at a scan rate of 1 Hz, to generate triboelectric charges. In the experiments of Figs 3 and 4, we applied a tip bias from −10 to 10 V on each sample with the conditions otherwise the same. The Kelvin probe force microscopy (KPFM) maps have been obtained with the tip biased by an AC voltage having amplitude and frequency equal to 2 V and 17 kHz, respectively. In addition, all AFM-based measurements progressed under the same conditions (temperature = 21 °C, relative humidity = 25 ∼ 30%).

**Data availability.** The data that support the findings of this study are available from the corresponding authors upon reasonable request.

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

## Acknowledgements

This project was financially supported by the Basic Science Research Program (2015R1A2A1A05001851) through the National Research Foundation (NRF) of Korea Grant funded by the Ministry of Science, ICT & Future Planning. SKKU has filed a patent related to this work.

## Author contributions

S.K., T.Y.K., S.-W.K and C.F. designed and conceptualized the project. S.K., T.Y.K., K.H.L., S.-W.K. and C.F. designed the experiments. K.H.L. and T.-H.K. synthesized graphene samples in the CVD process and fabricated devices. T.Y.K., R.H. and F.A.C. performed simulations. S.K., T.Y.K., S.K.K. and R.H. conducted the AFM and KPFM experiments and characterization. S.K., T.Y.K., K.H.L., R.H. and F.A.C. analysed the data. C.F. and S.-W.K. supervised the overall conception and design of this project. All authors contributed to the writing of the paper.

## Additional information

**Competing interests:** S.K., T.Y.K, K.H.L, S.K.K, C.F. and S.-W.K are co-inventors of the patent entitled 'Method for controlling electrical property of 2D material using triboelectrification' patent number: 10-1557246, South Korea. The remaining authors declare no competing financial interests.

