## [Peer Review File · Nature Communications]

Reviewers' Comments:

Reviewer #1 (Remarks to the Author)

The authors reported a way to build a special FET devices design with the tunneling-triboelectrification charges as the gate. This would be a very great idea if this design can come true in the very-large-scale integration (VLSI). However, the paper is still very mature at the current stage, they only investigated the charge distribution, time-dependent, and polarity control with different biasing AFM tip. Most of these research have been demonstrated before by other groups, as shown below. Furthermore, the authors do not shown any work in the device using the tribocharges as the ghost gate. Lastly, how to resolve the tribo-charge migration, and time-dependent-variation are key issues in the true applications.

In summary, I do not recommend this paper for Nature communication.

Reviewer #2 (Remarks to the Author)

The manuscript of S. Kim et al. presents an experimental study on supported chemical vapor deposition (CVD) graphene with local sudo-gates defined by tunneling triboelectrification (TT). The use of TT for graphene is novel and could potentially open a new avenue for new classes of devices and applications. The study is comprehensive, data is well presented, and the manuscript is well written and could be published in its current form.

Reviewer #3 (Remarks to the Author)

This manuscript presents an interesting concept of creating rewritable ghost floating gates by trapping the triboelectric charges between the air-SiO₂ interfaces, which is beneath the graphene deposited on a SiO₂/Si substrate. These charges are generated by friction between a Pt-coated atomic force microscope (AFM) and the few-layer graphene, and can be preserved for a long time. Using the tunneling-triboelectrification, the creation, modification and erasion of triboelectric charges could be realized by manipulating the AFMs. However, a few points are not clearly demonstrated.

1. According to the manuscript, in the beginning, the surface of the graphene (a 1.5×1.5 μm² area) has been rubbed by an AFM and the corresponding surface potential of the area has also been measured using Kelvin probe force microscopy (KPFM). The potential difference between the unrubbed and rubbed regions, as well as the average surface potential changes which result from rubbing has been used to prove that the tunneling-triboelectrification does happen. However, the potential difference between the unrubbed and rubbed region could be reasonably ascribed to the friction of the graphene layer and the SiO₂ layer, since a force is applied upon the graphene by an AFM. When there is fewer layer of graphene (e.g. 1 layer), the relative sliding between the two layers is more likely to happen. As the number of layers of graphene increases, the graphene is less likely to slide along the SiO₂ surface. Therefore, the amount of the triboelectric charges generated would decrease. As a result, the data and figures in the manuscript are insufficient to rule out the possibility of the sliding-induced triboelectric charges and the working mechanism proposed for potential change is problematic. In order to clearly show the existence of such claimed tunneling-triboelectrification, more precise analysis and experiments are needed to be performed.

2. Since the author has pointed out that the CVD graphene may have high defect density that the monolayer exfoliated graphene (MEG, Figure S5) does, there is a possibility that those triboelectric charges induced by the AFM could be trapped by these defects or impurities, which means those charges may not store at the air-SiO₂ interface. Therefore, if the author wants to prove that the triboelectric charges do tunnel through graphene and is injected at the interface between air and SiO₂ instead of just being trapped at the surface of the highly defective few-layer graphene, more experimental results should be provided in the figures and the manuscript.

3. As a key parameter, the current flowing in the graphene should be modulated by the so-called ghost floating gates. But in Figure 4b, only the relationship between the normalized resistance R/R_0 and the tunneling-triboelectric charge density is depicted. The author should provide more data to demonstrate the dependence of the current on the tip voltage or the ghost floating gates.

Reviewer #1 – Comment A

The authors reported a way to build a special FET devices design with the tunneling-triboelectrification charges as the gate. This would be a very great idea if this design can come true in the very-large-scale integration (VLSI).

Response

We would like to thank the referee for the critical reading and valuable comments which have helped us to make the manuscript more clear. We are also very thankful for the extremely positive comments on the potential of the proposed concepts in view of VLSI integration.

Reviewer #1 – Comment B

However, the paper is still very mature at the current stage, they only investigated the charge distribution, time-dependent, and polarity control with different biasing AFM tip. Most of these research have been demonstrated before by other groups, as shown below.

Response

We could not find references to previous works in the reviewer response. However, we think the reviewer may refer to previous papers (on the nanoscale control of charges on an insulating substrate, e.g. SiO₂, by AFM-tip induced triboelectrification) which were already included in the references list ([12, 13] in the main text). However, our work is not restricted to the control of the charge distribution/polarity and to the analysis of the decay of the charges after triboelectrification of an insulating substrate with an AFM tip.

In comparison with literature, our manuscript contains both scientific and technological advances and, in particular, we have:

- a) *found that when the AFM tip generates triboelectric charges on a 2D material (graphene), part of the charges can tunnel through the 2D material and, then, localize on the underlying insulator and, there, effectively control the properties of the 2D material (we have referred to this unreported phenomenon as tunneling-triboelectrification)*
- b) *experimentally verified that the decay of tunneling-triboelectric charges is over 100 times slower than the decay of conventional triboelectric charges on an insulating substrate (we found an exceptionally long time constant, more than two orders of magnitude longer than for conventional triboelectrification; such a long lifetime may offer important advantages for practical applications, e.g. for memories), see Fig. 1e*
- c) *investigated the physics of tunneling-triboelectrification and determined an accurate and quantitative model (e.g. see the equivalent circuit, Fig. 2e) which justifies the presence of two distinct time constants and explains why most of the charges localize on the air-gap capacitor rather than on the oxide capacitor and, therefore, survive for very long times (see the main text, Supplementary Sections 3.1, 3.2, 3.3 and Supporting Fig. S8-S10)*

- d) *proposed (with full experimental demonstration, see Fig. 3-5 and Fig. S11) that after tunneling, the stored charges can act as immaterial (charges-only), re-writable ghost floating gates which can be created, modified or destroyed at will, in striking contrast with conventional gates which, after manufacturing, are unchangeable*
- e) *proposed (with full experimental demonstration, see Fig. 3-5 and Fig. S11) that the re-writable floating gates can induce re-writable p and n regions in the 2D material; based on these results, we have envisioned re-writable field-effect transistors (with floating gates and, therefore, with restrictions for circuit design due to the physical inaccessibility of the gates) as well as other re-writable electronic/optoelectronic devices (e.g. diodes and bipolar junction transistors, with the possibility to physically contact all the terminals); most remarkably, the re-writable floating gates, p and n regions can be defined by atomic force microscopes (AFMs) with their typically extraordinary resolution; to the best of our knowledge, this is the first report that gates, p and n regions can be written, erased and re-written (eventually, at a different location and with a different shape)*
- f) *verified the existence of a threshold current for erasing the stored charges (see Fig. S11), which was impossible with triboelectrification on an insulating substrate only*
- g) *experimentally demonstrated that the re-writable ghost gates can effectively control the charge transport in the 2D material (Fig. 4a, 4b)*
- h) *experimentally demonstrated several re-writable floating gates, p/p^+ and p/n^+ junctions with resolution as small as 200 nm, limited by available instrumentation, see Fig. 5*
- i) *discussed the CMOS compatibility of devices defined by tunneling-triboelectrification*
- j) *discussed the possibly easy co-integration, by tunneling-triboelectrification, on the same substrate of devices using different 2D materials (e.g. graphene and MoS_2)*
- k) *proposed that, since AFMs can be fully integrated in a single chip, re-writable time-variant 2D devices (hardware) can be defined on-demand, in striking contrast with all existing electronic devices.*

Reviewer #1 – Comment C

Furthermore, the authors do not shown any work in the device using the tribocharges as the ghost gate.

Response

We already demonstrated, in practical devices, that key properties of 2D materials (e.g. electrical conductivity, polarity and concentrations of free charges) can be accurately controlled by tunneling-triboelectrification.

In particular, we described time-variant, deep sub-micron p/p^+ and p/n^+ junctions and rewritable floating gates (Fig. 5a, 5b) characterized by Kelvin Probe Force Microscopy; as shown in Fig. 5,

our experiments unambiguously demonstrate the creation of time-variant deep sub-micron p/p+ and p/n+ junctions defined by selectively rubbing arbitrary graphene regions with the AFM tip, thus demonstrating that the underlying rewritable deep sub-micron floating gates can control the polarity and concentration of free charges inside graphene (the same result is also illustrated by the two low-resistances regions in Fig. 4b, which further confirm the ability to (reversibly) switch from n-type to p-type charge transport and vice versa).

We emphasize that, once these possibilities (namely, to create p/p+ and p/n+ junctions and control the polarity and concentration of free charges) are clearly demonstrated, it may become obvious, for instance to experienced electronic engineers, how to design many types of electronic devices.

In fact, as a first example, we have already reported a resistor whose resistance can be controlled by tunneling-triboelectrification (Fig. 4), along with its full electrical characterization. These results, in perfect agreement with theoretical predictions, already constitute a complete device-level demonstration of tunneling-triboelectrification, because tunable resistors are crucial components in several electronic circuits (e.g. amplifiers with tunable gains, oscillators with automatic gain control, tunable voltage/current references/ bandgap circuits, ...).

*In order to illustrate more clearly that the resistor whose resistance can be controlled by tunneling-triboelectrification already represents a practical device (as well as to respond to the **Reviewer #3 – Comment C**), we have added some discussion and more data (which previously were simply summarized in Fig. 4b) to the Supporting Information, as follows.*

Section 4. Control of CVD graphene devices by ghost floating gates

4.1 Resistance control on CVD graphene resistor

Figure S12 shows the resistance reduction induced on the 4-contacts 1L CVD graphene resistor shown in Fig. 4. In practice, as schematically shown in Fig. S12a, we performed 4-wires measurements by injecting a 100 nA current through the force electrodes F_1 and F_2 and measuring the voltage difference across the inner sense electrodes S_1 and S_2 (the instrumentation amplifier IA has negligible input currents and, therefore, first, the entire I_0 current flows through the graphene resistor R_G , and, second, the voltage across the input terminals of the instrumentation amplifier is exactly the same as the voltage across R_G). For instance, Fig. S12b-d show the voltages across R_G (i.e. $100 \text{ nA} \times R_G$) before (higher value of R_G and, therefore, of the voltage difference across R_G) and after rubbing the middle area (yellow dashed box, between the sense electrodes S_1 and S_2 in Fig. 4a) with an AFM-Pt tip biased at -10 V (b), -5 V (c), and 0 V (d), respectively. As evident, the tunneling-triboelectric charges act as ghost floating gates and electrostatically reduce the normalized resistance R/R_0 (where R is the resistance of graphene after rubbing and R_0 is the resistance before rubbing). These results already constitute a device-level demonstration of tunneling-triboelectrification (e.g. tunable resistors are useful in tunable amplifiers, automatic gain control circuits, tunable voltage/current references, etc.).

Figure S12 | Resistance reduction of graphene by tunneling-triboelectrification. a, Schematic representation of 4-contacts 1L CVD graphene resistor and of the 4-wires measurement technique. **b-d,** Voltages measured across the sense terminals S_1 and S_2 with $I_0 = 100$ nA before and after AFM tunneling-triboelectrification performed with tip bias voltages of (b) -10 V, (c) -5 V, (d) 0 V.

Finally, in the revised manuscript, as another device-level demonstration, we have now shown that the current transport characteristics of a graphene FET (e.g. Dirac point) can be controllably changed by tunneling-triboelectrification, as follows.

4.2 Dirac point shift of graphene FET by tunneling-triboelectrification

In addition to the control of graphene resistivity by tunneling-triboelectrification (Fig. 4 and S12), we also verified that tunneling-triboelectrification allows to control the Dirac point of graphene in FET devices (Fig. S13). In practice, we measured the drain-to-source current of a graphene resistor as a function of the back gate voltage both before and after rubbing p-doped graphene with an AFM tip biased at +10 V. As described in Fig. S13b, after tunneling-triboelectrification, the Dirac point of the graphene resistor shifts to the left and, with a zero gate voltage, the effective doping turns from p-type to n-type. In fact, consistently with our theoretical discussions, first, positive charges are generated in graphene by triboelectrification with the positively biased AFM tip and, second, part of these positive charges tunnel through the air gap and can effectively turn the naturally p-type graphene into n-type (Fig. S13a). This situation is exactly the same found in MOS (metal-oxide-semiconductor) capacitors where the presence of charges on the metal gate induces opposite charges on the semiconductor and, therefore, travelling through the semiconductor at the semiconductor-oxide interface, going from the region underneath the gate to the surrounding regions of the semiconductor, there are potential differences (due to the gate charges) which may be well preserved for very long times. This is exactly analogous to our case, the difference being that our gate is floating, immaterial, and re-writable (or, equivalently, time variant). In conclusion,

Figure S13b confirms that tunneling-triboelectrification can effectively control the current transport characteristics of 2D FET devices.

We mention that Figure S13b also provides an additional confirmation that the positive charges induced on graphene by rubbing with the positively biased AFM tip (+10 V) tunnel through the air gap and localize at the SiO₂-air interface; in fact, such localized positive charges electrostatically attract free charges of the opposite type (negative) in graphene (i.e. turn graphene from p-type to n-type doping), similar to what would happen with positive charges localized on an hypothetical gate underneath the air-gap.

Figure S13 | Control of the Dirac point by tunneling-triboelectrification. a, Schematic image of the device structure and of the ghost floating gate. **b**, I_d-V_g curves before (black) and after (blue) tunneling-triboelectrification.

Reviewer #1 – Comment D

Lastly, how to resolve the tribo-charge migration, and time-dependent-variation are key issues in the true applications.

Response

Clearly, the main goal of this manuscript is not to discuss a commercial application of tunneling-triboelectrification (which may, of course, take some time), but:

- present an unreported physical effect (tunneling-triboelectrification),
- discuss the fundamental physics of tunneling-triboelectrification
- suggest how tunneling-triboelectrification can open the way to advanced devices that, by far, may not be doable with existing technologies
- provide device-level demonstrations of tunneling triboelectrification (Fig. 4, 5 and, in the revised manuscript, Fig. S12, S13)

We also agree that the decay of charges and the resulting time-dependent variations may be key issues for commercial applications.

However, from this perspective, our work already is a remarkable advance in comparison with the state of the art. In fact, the time decay of tunneling triboelectric charges is more than 2 orders of magnitude longer than the time decay of conventional triboelectrification charges on an insulating substrate. As an example, we found a time constant for the charges decay around 278 hours (i.e.

more than 11 days), to be compared with time constants around 1 hour for conventional triboelectrification.

Such a long lifetime may be insufficient for some applications but may already be enough in some cases, as our experiments consistently confirm that charges can be well preserved for many days.

Moreover, most importantly, we have already discussed how this issue can be fully overcome with existing technologies: as mentioned in the main text, it is already possible to integrate many AFM tips and also many different 2D devices on a single chip in order to make re-writable time-variant 2D devices which can be defined on-demand. In order to make this discussion more clear, we have added in the revised manuscript the following sentence:

This approach would be similar to the memory refresh process (i.e. periodically reading and immediately rewriting the same information on capacitive memories in order to counteract the degradation of information due to leakage currents) routinely used in dynamic random-access memories (DRAM).

Reviewer #1 – Comment E

In summary, I do not recommend this paper for Nature communication.

Response

Again, we are very thankful to the referee for the critical reading and valuable comments.

We also hope that our revisions and responses have improved and further clarified our manuscript.

Reviewer #2 – Comment

The manuscript of S. Kim et al. presents an experimental study on supported chemical vapor deposition (CVD) graphene with local sudo-gates defined by tunneling triboelectrification (TT). The use of TT for graphene is novel and could potentially open a new avenue for new classes of devices and applications. The study is comprehensive, data is well presented, and the manuscript is well written and could be published in its current form.

Response

We are very thankful to the referee for the critical reading and the very positive comments.

Reviewer #3 – Comment A

This manuscript presents an interesting concept of creating rewritable ghost floating gates by trapping the triboelectric charges between the air-SiO₂ interfaces, which is beneath the graphene deposited on a SiO₂/Si substrate. These charges are generated by friction between a Pt-coated atomic force microscope (AFM) and the few-layer graphene, and can be preserved for a long time. Using the tunneling-triboelectrification, the creation, modification and erasure of triboelectric

charges could be realized by manipulating the AFMs. However, a few points are not clearly demonstrated.

1. According to the manuscript, in the beginning, the surface of the graphene (a $1.5 \times 1.5 \text{ } \mu\text{m}^2$ area) has been rubbed by an AFM and the corresponding surface potential of the area has also been measured using Kelvin probe force microscopy (KPFM). The potential difference between the unrubbed and rubbed regions, as well as the average surface potential changes which result from rubbing has been used to prove that the tunneling-triboelectrification does happen. However, the potential difference between the unrubbed and rubbed region could be reasonably ascribed to the friction of the graphene layer and the SiO_2 layer, since a force is applied upon the graphene by an AFM. When there is fewer layer of graphene (e.g. 1 layer), the relative sliding between the two layers is more likely to happen. As the number of layers of graphene increases, the graphene is less likely to slide along the SiO_2 surface. Therefore, the amount of the triboelectric charges generated would decrease. As a result, the data and figures in the manuscript are insufficient to rule out the possibility of the sliding-induced triboelectric charges and the working mechanism proposed for potential change is problematic. In order to clearly show the existence of such claimed tunneling-triboelectrification, more precise analysis and experiments are needed to be performed.

Response

We would like to thank the referee for the critical reading, positive and valuable comments which have helped us to make the manuscript more clear.

We are especially thankful for suggesting that the triboelectric charges might be generated by the rubbing between graphene and the insulating substrate. Therefore, we agree this possibility must be explicitly discussed in the manuscript.

Accordingly, we have now modified and expanded the Section 2 in the Supporting Information and have given the reasons to rule out this possibility. The revised Section 2 of SI relative to this specific comment is as follows.

Section 2. Additional experiments on the tunneling-triboelectrification mechanism

2.1 Tunneling-triboelectrification by friction between the AFM tip and graphene

The potential difference between the unrubbed and rubbed region could in principle be ascribed to the friction, induced by the AFM tip, between the graphene layer and the SiO_2 layer. However, this possibility can be ruled out for the following reasons.

First, we could not obtain detectable potential differences between the rubbed and the unrubbed parts when using mechanically exfoliated graphene (MEG, see Fig. 2b and 2c), consistently with the tunneling mechanism (mechanically exfoliated graphene is clearly less defective than CVD graphene). By contrast, the hypothetical friction of mechanically exfoliated graphene with SiO_2 would likely give comparable (at least, for the orders of magnitude) results as the hypothetical friction of CVD graphene with SiO_2 .

Second, the analysis of the triboelectric series for SiO_2 , Pt and graphene reveals that our results may not be explained by the friction of graphene and SiO_2 . In order to determine the relative positions of SiO_2 and Pt in the triboelectric series, we rubbed SiO_2 (300 nm) over a $2 \times 2 \text{ } \mu\text{m}^2$ area using a grounded Pt-coated AFM tip in contact mode and measured the surface potential ($5 \times 5 \text{ } \mu\text{m}^2$) both before and after triboelectrification, similar to the experiments shown in Fig. 1. Figure S3a shows

the uniform surface potential of SiO₂ before rubbing. Figure S3b shows that, after rubbing, the surface potential of the rubbed region decreased contrastively to results in Fig. 1c which shows an increased surface potential in the rubbed region. As a consequence, Pt is positive with respect to SiO₂ in the triboelectric series. In case of friction between Pt and graphene (Fig. S3d), the sign of the triboelectric charges is determined by the work functions (Φ) of the two materials. In practice, after friction, electrons transfer from graphene ($\Phi_{\text{Graphene}} \approx 4.5$ eV) to Pt ($\Phi_{\text{Pt}} \approx 5.9$ eV) and then, graphene becomes positive and Pt becomes negative. The resulting triboelectric series of SiO₂, Pt, and graphene is schematically described in Fig. S3f, so that, in the hypothetical case of friction between SiO₂ and graphene, SiO₂ would become negative and graphene positive (Fig. S3e), contrary to our experimental results.

In conclusion, both the absence of detectable effects when using mechanically exfoliated graphene and the triboelectric series of SiO₂, Pt, and graphene confirm that the localization of charges on the insulator underneath graphene is not induced by friction between graphene and SiO₂, but is determined by the tunneling of triboelectric charges generated by friction between the Pt-coated AFM tip and graphene.

Figure S3 | Triboelectric series of SiO₂, Pt and graphene. **a**, KPFM image of SiO₂ before and **b**, after rubbing with Pt-coated AFM tip. **c-e**, Schematic representation of the triboelectric charge transfer processes in the cases of **(c)** SiO₂/Pt, **(d)** Pt/graphene and **(e)** SiO₂/graphene. **f**, Triboelectric series of SiO₂, Pt and graphene.

Reviewer #3 – Comment B

2. Since the author has pointed out that the CVD graphene may have high defect density that the monolayer exfoliated graphene (MEG, Figure S5) does, there is a possibility that those triboelectric charges induced by the AFM could be trapped by these defects or impurities, which means those charges may not store at the air-SiO₂ interface. Therefore, if the author wants to prove that the triboelectric charges do tunnel through graphene and is injected at the interface between air and SiO₂ instead of just being trapped at the surface of the highly defective few-layer graphene, more experimental results should be provided in the figures and the manuscript.

Response

We are very thankful for this comment.

We agree this is an important point and have accordingly modified the Section 2 in the Supporting Information for explicitly discussing why this possibility can be ruled out.

The first reason is the very long decay time of the charges, which would be impossible if the charges were trapped in graphene because of its good electrical conductivity (see below, Section 2.2 of the revised SI).

Second, we consistently found detectable potential differences between the rubbed and unrubbed parts if and only if there was an insulator underneath graphene where charges can localize and stay for long times (see below, Section 2.3 of the revised SI).

Finally, the same experiment we have included as an additional device level demonstration of tunneling-triboelectrification (i.e. shift of the Dirac point) also confirms that the charges are trapped in an insulator separated from graphene. In fact, the triboelectric positive charges (created by rubbing graphene with the Pt-AFM tip biased at the highest positive voltage, +10 V) do not result in an enhanced p-type graphene doping but in n-type doping because the positive charges tunnel through the air gap, localize at the SiO₂-air interface and electrostatically attract free charges of the opposite type (negative) in graphene, i.e. turn graphene from p-type to n-type doping (see below, the last paragraph of Section 4.2 of the revised SI).

The revised Sections 2 and 4 of SI relative to this specific comment is as follows.

2.2 Localization of charges at the air-SiO₂ interface

In principle, the triboelectric charges might also be trapped in the defects or impurities of CVD graphene rather than being stored at the air-SiO₂ interface. In fact, this is possible during a very short transient, but this mechanism may not justify the very slow decay of the potential difference between the rubbed and unrubbed graphene areas (e.g. see Fig. 1d, 1e and S4). Additional charges trapped in graphene defects or impurities would create potential differences in graphene, but, since CVD graphene is not insulating, these potential differences would immediately result in currents which would tend to make graphene equipotential. In CVD graphene, different from insulators, these processes would be extremely fast (i.e. in conductors or semiconductors, significant potential differences may not be maintained for long times in absence of an external perturbation). In other words, after a very fast transient, potential differences across the CVD graphene (similar to other conductors or semiconductors) would quickly disappear.

By contrast, the existence, for very long times (e.g. many days), of potential differences in graphene is easily explained by the presence of electric charges localized on an insulator underneath graphene, similar to MOS (metal-oxide-semiconductor) capacitors, the difference being that our gate is floating, immaterial, and re-writable (i.e. time variant). The localization of charges at the air-

SiO₂ interface is also confirmed by the accuracy of the equivalent circuit shown in Fig. 2d, which gives reasons of the presence of a shorter time constant (associated to the discharge of the oxide capacitor) and of a longer time constant (associated to the discharge of the air-gap capacitor) as well as of the higher magnitude of the slow-decay term (almost all the charges localize on the air-gap capacitor rather than on the SiO₂ capacitor, see main text). We mention that the localization of the charges at the air-SiO₂ interface is also confirmed by the Dirac point shift (Supplementary Section 4 and Fig. S13).

2.3 Triboelectrification of graphene on different substrates

As an additional confirmation, in order to further verify that charges are trapped at the air-gap SiO₂ interface, we also carried out identical experiments with conductive metal substrates. The CVD graphene sheet was transferred on copper (Cu) substrates treated by HF to remove the native oxide layer (Fig. S5). Then, the top surface of graphene was rubbed with the Pt-coated AFM tip and the surface potential was measured using KPFM. In contrast with the case of insulating substrates, triboelectric charges were not localized in the rubbed regions (white dashed square) but spread to the whole region (Fig. S5c), thus confirming that the presence of an insulator under graphene is crucial.

Figure S5 | Triboelectrification of graphene on metal substrate. **a**, Schematic image of graphene transferred on metal (Cu) substrate. **b**, KPFM image of graphene on metal substrate before rubbing and **c**, after rubbing with Pt-coated AFM tip.

Moreover, we also repeated the same experiments with CVD graphene deposited on other insulating substrates such as mica and Al₂O₃ (see Fig. S6) and, similar to the case of CVD graphene on SiO₂, found that charges, after tunneling through CVD graphene, were trapped at the interface between air-gap and SiO₂.

Figure S6 | Tunneling-triboelectrification of graphene on mica and on Al_2O_3 . a, b, KPFM images, after tunneling-triboelectrification, of CVD graphene deposited on (a) Mica and CVD graphene deposited on (b) Al_2O_3 . c, d, Cross-sectional profiles along the black dashed lines for graphene on (c) Mica and graphene on (d) Al_2O_3 after 0 min and 12 hours.

4.2 Dirac point shift of graphene FET by tunneling-triboelectrification

In addition to the control of graphene resistivity by tunneling-triboelectrification (Fig. 4 and S12), we also verified that tunneling-triboelectrification allows to control the Dirac point of graphene in FET devices (Fig. S13). In practice, we measured the drain-to-source current of a graphene resistor as a function of the back gate voltage both before and after rubbing p-doped graphene with an AFM tip biased at +10 V. As described in Fig. S13b, after tunneling-triboelectrification, the Dirac point of the graphene resistor shifts to the left and, with a zero gate voltage, the effective doping turns from p-type to n-type. In fact, consistently with our theoretical discussions, first, positive charges are generated in graphene by triboelectrification with the positively biased AFM tip and, second, part of these positive charges tunnel through the air gap and can effectively turn the naturally p-type graphene into n-type (Fig. S13a). This situation is exactly the same found in MOS (metal-oxide-semiconductor) capacitors where the presence of charges on the metal gate induces opposite charges on the semiconductor and, therefore, travelling through the semiconductor at the semiconductor-oxide interface, going from the region underneath the gate to the surrounding regions of the semiconductor, there are potential differences (due to the gate charges) which may be well preserved for very long times. This is exactly analogous to our case, the difference being that our gate is floating, immaterial, and re-writable (or, equivalently, time variant). In conclusion, Figure S13b confirms that tunneling-triboelectrification can effectively control the current transport characteristics of 2D FET devices.

We mention that Figure S13b also provides an additional confirmation that the positive charges induced on graphene by rubbing with the positively biased AFM tip (+10 V) tunnel through the air gap and localize at the SiO_2 -air interface; in fact, such localized positive charges electrostatically attract free charges of the opposite type (negative) in graphene (i.e. turn graphene from p-type to n-

type doping), similar to what would happen with positive charges localized on an hypothetical gate underneath the air-gap.

Figure S13 | Control of the Dirac point by tunneling-triboelectrification. a, Schematic image of the device structure and of the ghost floating gate. **b**, I_d - V_g curves before (black) and after (blue) tunneling-triboelectrification.

Reviewer #3 – Comment C

3. As a key parameter, the current flowing in the graphene should be modulated by the so-called ghost floating gates. But in Figure 4b, only the relationship between the normalized resistance R/R_0 and the tunneling-triboelectric charge density is depicted. The author should provide more data to demonstrate the dependence of the current on the tip voltage or the ghost floating gates.

Response

We are very thankful for this comment.

The data in Fig. 4b have been derived by measurements of currents flowing through the resistor shown in Fig. 4a; for conciseness, we have given directly the measurement of the normalized resistance R/R_0 as a function of the localized charge density. However, we agree that the original measurements may also be informative and have therefore added Fig. S12 and the correspondent discussion in the Supporting Information as follows.

Section 4. Control of CVD graphene devices by ghost floating gates

4.1 Resistance control on CVD graphene resistor

Figure S12 shows the resistance reduction induced on the 4-contacts 1L CVD graphene resistor shown in Fig. 4. In practice, as schematically shown in Fig. S12a, we performed 4-wires measurements by injecting a 100 nA current through the force electrodes F_1 and F_2 and measuring the voltage difference across the inner sense electrodes S_1 and S_2 (the instrumentation amplifier IA has negligible input currents and, therefore, first, the entire I_0 current flows through the graphene resistor R_G , and, second, the voltage across the input terminals of the instrumentation amplifier is exactly the same as the voltage across R_G). For instance, Fig. S12b-d show the voltages across R_G (i.e. $100 \text{ nA} \times R_G$) before (higher value of R_G and, therefore, of the voltage difference across R_G) and after rubbing the middle area (yellow dashed box, between the sense electrodes S_1 and S_2 in Fig.

4a) with an AFM-Pt tip biased at -10 V (b), -5 V (c), and 0 V (d), respectively. As evident, the tunneling-triboelectric charges act as ghost floating gates and electrostatically reduce the normalized resistance R/R_0 (where R is the resistance of graphene after rubbing and R_0 is the resistance before rubbing). These results already constitute a device-level demonstration of tunneling-triboelectrification (e.g. tunable resistors are useful in tunable amplifiers, automatic gain control circuits, tunable voltage/current references, etc.).

Figure S12 | Resistance reduction of graphene by tunneling-triboelectrification. a, Schematic representation of 4-contacts 1L CVD graphene resistor and of the 4-wires measurement technique. **b-d,** Voltages measured across the sense terminals S_1 and S_2 with $I_0 = 100$ nA before and after AFM tunneling-triboelectrification performed with tip bias voltages of (b) -10 V, (c) -5 V, (d) 0 V.

Reviewers' Comments:

Reviewer #1:

Remarks to the Author:

I am satisfied with the revisions made by the authors, and the MS is suitable for publication now.

Reviewer #3:

Remarks to the Author:

The authors has made some revisions according to the reviewer's concern, and some of the additions do make the explanations more persuasive. For example, additional experiments has been provided to clarify the tunneling-triboelectrification mechanism and rule out the possibility of friction between the graphene layer and the SiO₂ layer, which may cause the same effect. Also, the researchers used experiments to verify that charges are trapped at the air-gap SiO₂ interface, instead of being trapped by defects or impurities of CVD graphene.

However, as the so-called "ghost floating gates" are critical to this manuscript, representative applications should be provided to demonstrate how such gates can modulate device-level electronics. In fact, the demonstration of the graphene resistor (both the variable resistor experiment and the Dirac point shift of the graphene) is insufficient to provide a device-level demonstration. Furthermore, this manuscript do not show any data of the transfer curve and the output curve of the graphene-based FET devices. In Figure S13b, the author did provide I_d - V_g curves before and after tunneling-triboelectrification. However, here the V_g is not the "true gate voltage". It is the localized positive charges that play the role of the "true gate voltage" to attract free charges of the opposite type in graphene. Therefore, there is no direct and quantitative analysis about the relationship between the "ghost floating voltage" and the drain-source current.

In summary, I do not recommend this paper for Nature Communications.

For clarity, we refer to the new Reviewer #3 comments as to Comments A2, B2,...

Reviewer #3 – Comment A2

The authors has made some revisions according to the reviewer's concern, and some of the additions do make the explanations more persuasive. For example, additional experiments has been provided to clarify the tunneling-triboelectrification mechanism and rule out the possibility of friction between the graphene layer and the SiO₂ layer, which may cause the same effect. Also, the researchers used experiments to verify that charges are trapped at the air-gap SiO₂ interface, instead of being trapped by defects or impurities of CVD graphene.

Response

We would like to thank the referee for both the critical reading and the positive comments on our revisions and additional experiments.

Reviewer #3 – Comment B2

However, as the so-called “ghost floating gates” are critical to this manuscript, representative applications should be provided to demonstrate how such gates can modulate device-level electronics. In fact, the demonstration of the graphene resistor (both the variable resistor experiment and the Dirac point shift of the graphene) is insufficient to provide a device-level demonstration.

Response

We have already presented representative applications of the ghost floating gate concept. Though, of course, the commercialization of devices based on tunneling triboelectrification may require more time, we have already demonstrated some applications at device-level (i.e. we have shown functional devices based on tunneling triboelectrification).

As a first application, we have shown, with device-level demonstration, high-density memories which store information in ghost virtual gates; these memories have a decay time about 2 orders of magnitude longer than the corresponding memories using conventional triboelectrification.

As a second application, we have shown, with device-level demonstration, time-variant resistance control. The device shown in Fig. 4 is a functional resistor whose resistance can be controlled by tunneling-triboelectrification (similar devices may be used in tunable amplifiers, automatic gain control, tunable voltage/current references, instrumentation amplifiers with variable gain, etc.).

As a third application, we have shown, with device-level demonstration, a graphene FET whose Dirac point can be controlled by tunneling triboelectrification. In practice, the device shown in Fig. S13a, comprises the channel, with its two terminals, and the back gate (which extends from the

metal contact to the high-doped silicon) and constitutes a functional graphene FET device whose current transport properties can be effectively controlled by tunneling triboelectrification.

Finally, we also mention that in our previous response to **Reviewer #1** we also discussed practical applications of tunneling-triboelectrification; for completeness, we report our previous responses to the **Reviewer #1 (Comments C and D)** at the end of the present response.

In conclusion, we believe our results experimentally confirm, with device-level demonstrations, that tunneling-triboelectrification can open the way to advanced devices that, by far, may not be doable with existing technologies.

Reviewer #3 – Comment C2

Furthermore, this manuscript do not show any data of the transfer curve and the output curve of the graphene-based FET devices. In Figure S13b, the author did provide I_d - V_g curves before and after tunneling-triboelectrification. However, here the V_g is not the “true gate voltage”. It is the localized positive charges that play the role of the “true gate voltage” to attract free charges of the opposite type in graphene. Therefore, there is no direct and quantitative analysis about the relationship between the “ghost floating voltage” and the drain-source current.

In summary, I do not recommend this paper for Nature Communications.

Response

We are sorry but there is some misunderstanding (which could also explain the previous Comment B2 from the same Reviewer).

In order to make the text more clear, we now adopted the standard subscript convention (upper-case letters with upper-case subscripts refer to DC quantities, e.g. V_{BG} , V_{DS} , ...). With reference to figure S13b, we also noticed that we previously included a typo (the x-axis was labeled V_G instead of V_{BG}) and omitted to specify the value of the constant drain-to-source voltage ($V_{DS} = 0.01$ V) applied to the graphene FET during the measurements. We apologize and, of course, have now revised both figure S13b and text; we hope that everything is clear now.

The voltage on the x-axis in figure S13b is the true back-gate voltage (V_{BG}). In fact, figure S13b describes the most typical device-level demonstration of a graphene FET constituted by the channel, with two terminals, and a back gate as an additional terminal. In practice, we measured the current through the channel, with a given voltage ($V_{DS} = 0.01$ V) applied across the channel, as a function of the back-gate voltage. As a result, figure S13b demonstrates, at device level, that tunneling triboelectrification can shift the Dirac point of a functional graphene FET from the black curve (unrubbed) to the blue curve (after rubbing with AFM tip biased at +10V). Equivalently, figure S13b may also be seen as the transfer curve of a functional graphene FET, if we consider the back-gate voltage as the input signal and the drain-to-source current as the output signal. We have also added the sentence below in the Supplementary Note 4.2 to avoid confusion.

The I_{DS} - V_{BG} curves were measured under constant drain-to-source voltage ($V_{DS} = 0.01$ V).

We conclude that tunneling triboelectrification can control the electrical transport characteristics of functional graphene FETs.

For completeness, below we report our previous responses to the **Reviewer #1 Comments C and D**, modified by correcting the typo (x-axis in figure S13b is now correctly labeled V_{BG}), by revising the figure S13b caption, and adding the sentence:

“The I_{DS} - V_{BG} curves were measured under constant drain-to-source voltage ($V_{DS} = 0.01$ V).”

Reviewer #1 – Comment C

Furthermore, the authors do not shown any work in the device using the tribocharges as the ghost gate.

Response

We already demonstrated, in practical devices, that key properties of 2D materials (e.g. electrical conductivity, polarity and concentrations of free charges) can be accurately controlled by tunneling-triboelectrification.

In particular, we described time-variant, deep sub-micron p/p^+ and p/n^+ junctions and rewritable floating gates (Fig. 5a, 5b) characterized by Kelvin Probe Force Microscopy; as shown in Fig. 5, our experiments unambiguously demonstrate the creation of time-variant deep sub-micron p/p^+ and p/n^+ junctions defined by selectively rubbing arbitrary graphene regions with the AFM tip, thus demonstrating that the underlying rewritable deep sub-micron floating gates can control the polarity and concentration of free charges inside graphene (the same result is also illustrated by the two low-resistances regions in Fig. 4b, which further confirm the ability to (reversibly) switch from n-type to p-type charge transport and vice versa).

We emphasize that, once these possibilities (namely, to create p/p^+ and p/n^+ junctions and control the polarity and concentration of free charges) are clearly demonstrated, it may become obvious, for instance to experienced electronic engineers, how to design many types of electronic devices.

In fact, as a first example, we have already reported a resistor whose resistance can be controlled by tunneling-triboelectrification (Fig. 4), along with its full electrical characterization. These results, in perfect agreement with theoretical predictions, already constitute a complete device-level demonstration of tunneling-triboelectrification, because tunable resistors are crucial components in several electronic circuits (e.g. amplifiers with tunable gains, oscillators with automatic gain control, tunable voltage/current references/ bandgap circuits, ...).

In order to illustrate more clearly that the resistor whose resistance can be controlled by tunneling-triboelectrification already represents a practical device (as well as to respond to the **Reviewer #3 – Comment C**), we have added some discussion and more data (which previously were simply summarized in Fig. 4b) to the Supporting Information, as follows.

Note 4. Control of CVD graphene devices by ghost floating gates

4.1 Resistance control on CVD graphene resistor

Figure S12 shows the resistance reduction induced on the 4-contacts 1L CVD graphene resistor shown in Fig. 4. In practice, as schematically shown in Fig. S12a, we performed 4-wires measurements by injecting a 100 nA current through the force electrodes F_1 and F_2 and measuring the voltage difference across the inner sense electrodes S_1 and S_2 (the instrumentation amplifier IA has negligible input currents and, therefore, first, the entire I_0 current flows through the graphene resistor R_G , and, second, the voltage across the input terminals of the instrumentation amplifier is exactly the same as the voltage across R_G). For instance, Fig. S12b-d show the voltages across R_G (i.e. $100 \text{ nA} \times R_G$) before (higher value of R_G and, therefore, of the voltage difference across R_G) and after rubbing the middle area (yellow dashed box, between the sense electrodes S_1 and S_2 in Fig. 4a) with an AFM-Pt tip biased at -10 V (b), -5 V (c), and 0 V (d), respectively. As evident, the tunneling-triboelectric charges act as ghost floating gates and electrostatically reduce the normalized resistance R/R_0 (where R is the resistance of graphene after rubbing and R_0 is the resistance before rubbing). These results already constitute a device-level demonstration of tunneling-triboelectrification (e.g. tunable resistors are useful in tunable amplifiers, automatic gain control circuits, tunable voltage/current references, etc.).

Supplementary Figure S12 | Resistance reduction of graphene by tunneling-triboelectrification. (a) Schematic representation of 4-contacts 1L CVD graphene resistor and of the 4-wires measurement technique. (b)-(d) Voltages measured across the sense terminals S_1 and S_2 with $I_0 = 100 \text{ nA}$ before and after AFM tunneling-triboelectrification performed with tip bias voltages of (b) -10 V, (c) -5 V, (d) 0 V.

Finally, in the revised manuscript, as another device-level demonstration, we have now shown that the current transport characteristics of a graphene FET (e.g. Dirac point) can be controllably changed by tunneling-triboelectrification, as follows.

4.2 Dirac point shift of graphene FET by tunneling-triboelectrification

In addition to the control of graphene resistivity by tunneling-triboelectrification (Fig. 4 and Supplementary Fig. 12), we also verified that tunneling-triboelectrification allows to control the Dirac point of graphene in FET devices (Supplementary Fig. 13). In practice, we measured the drain-to-source current (I_{DS}) of a graphene resistor as a function of the back gate voltage (V_{BG}) both before and after rubbing p-doped graphene with an AFM tip biased at +10 V. The I_{DS} - V_{BG} curves were measured under constant drain-to-source voltage ($V_{DS} = 0.01$ V). As described in Supplementary Fig. 13b, after tunneling-triboelectrification, the Dirac point of the graphene resistor shifts to the left and, with a zero gate voltage, the effective doping turns from p-type to n-type. In fact, consistently with our theoretical discussions, first, positive charges are generated in graphene by triboelectrification with the positively biased AFM tip and, second, part of these positive charges tunnel through the air gap and can effectively turn the naturally p-type graphene into n-type (Supplementary Fig. 13a). This situation is exactly the same found in MOS (metal-oxide-semiconductor) capacitors where the presence of charges on the metal gate induces opposite charges on the semiconductor and, therefore, travelling through the semiconductor at the semiconductor-oxide interface, going from the region underneath the gate to the surrounding regions of the semiconductor, there are potential differences (due to the gate charges) which may be well preserved for very long times. This is exactly analogous to our case, the difference being that our gate is floating, immaterial, and re-writable (or, equivalently, time variant). In conclusion, Supplementary Fig. 13b confirms that tunneling-triboelectrification can effectively control the current transport characteristics of 2D FET devices.

We mention that Supplementary Fig. 13b also provides an additional confirmation that the positive charges induced on graphene by rubbing with the positively biased AFM tip (+10 V) tunnel through the air gap and localize at the SiO_2 -air interface; in fact, such localized positive charges electrostatically attract free charges of the opposite type (negative) in graphene (i.e. turn graphene from p-type to n-type doping), similar to what would happen with positive charges localized on an hypothetical gate underneath the air-gap.

Supplementary Figure 13 | Control of the Dirac point by tunneling-triboelectrification. (a) Schematic image of the device structure and of the ghost floating gate. (b) I_{DS} - V_{BG} curves before (black) and after (blue) tunneling-triboelectrification ($V_{DS} = 0.01$ V).

Reviewer #1 – Comment D

Lastly, how to resolve the tribo-charge migration, and time-dependent-variation are key issues in the true applications.

Response

Clearly, the main goal of this manuscript is not to discuss a commercial application of tunneling-triboelectrification (which may, of course, take some time), but:

- *present an unreported physical effect (tunneling-triboelectrification),*
- *discuss the fundamental physics of tunneling-triboelectrification*
- *suggest how tunneling-triboelectrification can open the way to advanced devices that, by far, may not be doable with existing technologies*
- *provide device-level demonstrations of tunneling triboelectrification (Fig. 4, 5 and, in the revised manuscript, Fig. S12, S13)*

We also agree that the decay of charges and the resulting time-dependent variations may be key issues for commercial applications.

However, from this perspective, our work already is a remarkable advance in comparison with the state of the art. In fact, the time decay of tunneling triboelectric charges is more than 2 orders of magnitude longer than the time decay of conventional triboelectrification charges on an insulating substrate. As an example, we found a time constant for the charges decay around 278 hours (i.e. more than 11 days), to be compared with time constants around 1 hour for conventional triboelectrification.

Such a long lifetime may be insufficient for some applications but may already be enough in some cases, as our experiments consistently confirm that charges can be well preserved for many days.

Moreover, most importantly, we have already discussed how this issue can be fully overcome with existing technologies: as mentioned in the main text, it is already possible to integrate many AFM tips and also many different 2D devices on a single chip in order to make re-writable time-variant 2D devices which can be defined on-demand. In order to make this discussion more clear, we have added in the revised manuscript the following sentence:

This approach would be similar to the memory refresh process (i.e. periodically reading and immediately rewriting the same information on capacitive memories in order to counteract the degradation of information due to leakage currents) routinely used in dynamic random-access memories (DRAM).

Reviewers' Comments:

Reviewer #3:

Remarks to the Author:

According to the whole manuscript of this work and the revisions made by the authors, I do not recommend this paper for publishing in Nature Communications.

In the first, since the authors have mentioned a device-level demonstration, that is a graphene FET, in order to clearly show the basic characteristics of the graphene transistor constructed for this device demonstration, both transfer and output behaviors need to be shown. But this manuscript didn't provide any information about the output curves of the transistor.

Secondly, the authors claims that the ghost floating gate is superior to the traditional one (e.g. the gate of the Si-based transistor) due to its immaterial, charge-only and changeable nature, which makes it extremely suitable for the 2D materials based device and practical use. However, during the whole experiment process, the ghost floating gates are created, modified and destroyed by the AFM tip. Compared with the very-large-scale integration (VLSI) of independent field-effect devices on a single chip, it might be much more complex and even impossible to realize in the 2D field-effect devices by the AFM tip.

After careful consideration, I do not recommend this manuscript for Nature Communications.

Reviewers' comments and *our responses (in italic)*

For clarity, we refer to the new Reviewer #3 comments as to Comments A3, B3,...

Reviewer #3 – Comment A3

According to the whole manuscript of this work and the revisions made by the authors, I do not recommend this paper for publishing in Nature Communications.

In the first, since the authors have mentioned a device-level demonstration, that is a graphene FET, in order to clearly show the basic characteristics of the graphene transistor constructed for this device demonstration, both transfer and output behaviors need to be shown. But this manuscript didn't provide any information about the output curves of the transistor.

Response

*We have discussed 3 distinct applications (not just “**mentioned a device-level demonstration**”).*

Moreover, in our opinion, the absence of a measurement could eventually justify a request to add the missing information, rather than rejection.

*The Reviewer #3 asks the measurement of the “**transfer and output behaviors**”.*

*We have shown the **transfer** characteristic (i.e. the “**transfer behavior**”) of the graphene FET (i.e. the $I_{DS}(V_{GS})$ relation, Fig. S13), which has been measured with a constant drain to source voltage $V_{DS,REF}(10\text{ mV})$, i.e. we have measured $I_{DS}(V_{GS})|_{V_{DS}=V_{DS,REF}}$. Moreover, in case of graphene FETs,*

*in contrast with conventional transistors (e.g. silicon BJTs or MOSFETs), **this measurement also gives information on the output characteristics** (i.e. the “**output behavior**”). For this reason, **we have not separately shown the $I_{DS}(V_{DS})$ relation, as it would be redundant and would provide no additional information, as is well known.** In fact, these characteristics are generally omitted in literature and only the $I_{DS}(V_{GS})$ characteristics are shown (same as we have done). Though this is so well known that is generally not discussed in papers, here we give the reasons for this choice.*

For conventional transistors (e.g. silicon BJTs or MOSFETs) the output characteristics (i.e. the $I_{DS}(V_{DS})$ curves) must be independently measured for determining the limits of the active region and the dynamic drain-to-source resistance r_{ds} , which are both crucial for analog circuit design. For instance, the output resistance of a simple current mirror is r_{ds} and the magnitude of the voltage gain of a common source amplifier is limited by $g_m r_{ds}$ (generally referred to as the intrinsic gain) where g_m is the transconductance. Moreover, the values of V_{DS} that ensure transistors are biased in the active region, so that r_{ds} is sufficiently high, must also be found from the output characteristics (e.g. for n-type MOSFETs biased in strong inversion, V_{DS} must be approximately larger than the overdrive voltage, i.e. $V_{GS} - V_{TH,N}$, but the real limits need careful characterization, especially in deep sub-micron processes). This, however, is not the case of graphene FETs.

*In fact, in striking contrast with conventional transistors (e.g. silicon BJTs or MOSFETs), due to the absence of a bandgap, graphene FETs may not be regarded as good voltage-controlled current-sources (i.e. there is no active region) and the $I_{DS}(V_{DS})$ curve, for a given V_{GS} , is simply linear. As a result, the **output characteristics can be easily derived from the transfer characteristics** (i.e. from*

*the $I_{DS}(V_{GS})$ characteristic) as $I_{DS}(V_{GS}, V_{DS}) = I_{DS}(V_{GS})|_{V_{DS}=V_{DS,REF}} * \frac{V_{DS}}{V_{DS,REF}}$.*

For this reason, the output characteristics of graphene FETs are often omitted even in papers which (unlikely the present manuscript) exclusively deal with graphene FETs (e.g. see the landmark paper [1] on top-gated graphene FETs or the first graphene chemical sensors with single-molecule resolution [2]).

For completeness, we mention that the above discussion applies to the typical case of relatively small V_{DS} values; in fact, when the magnitude of the V_{DS} voltage is sufficiently increased, the $I_{DS}(V_{DS})$ relation deviates from linearity and kinks appear due to the formation of an ambipolar channel [3]. Though these kinks may somewhat increase r_{ds} as it would be important for increasing the intrinsic gain of graphene amplifiers, the improvements are extremely limited so that, eventually, very sophisticated and complex circuit techniques are necessary for designing amplifiers [4]. As a consequence of such small r_{ds} , graphene FETs may not be seen as good voltage-controlled current sources and, therefore, may not be used for designing electronic circuits (e.g. voltage amplifiers or current mirrors) but may only be used for other applications, where the V_{DS} voltages are kept at reasonably small values and, therefore, the $I_{DS}(V_{DS})$ relation is linear.

In conclusion, since in most applications the V_{DS} voltages of graphene FETs are kept small, the output characteristics are not given (and, anyway, if desired, can be determined from the $I_{DS}(V_{GS})$ curves) and the $I_{DS}(V_{DS})$ relation is, explicitly or implicitly, assumed as linear. This is, for instance, an implicit choice in the so many papers where a graphene channel is described as a resistor. As just a few examples, the output characteristics have not been reported in papers describing top-gated graphene FETs [1], chemical sensors with single-molecule resolution [2], DNA sensors [5], humidity sensors [6], force sensors [7], microheaters [8] and strain sensors [9,10].

Reviewer #3 – Comment B3

Secondly, the authors claims that the ghost floating gate is superior to the traditional one (e.g. the gate of the Si-based transistor) due to its immaterial, charge-only and changeable nature, which makes it extremely suitable for the 2D materials based device and practical use. However, during the whole experiment process, the ghost floating gates are created, modified and destroyed by the AFM tip. Compared with the very-large-scale integration (VLSI) of independent field-effect devices on a single chip, it might be much more complex and even impossible to realize in the 2D field-effect devices by the AFM tip.

After careful consideration, I do not recommend this manuscript for Nature Communications.

Response

*This hypothetic (“**might be**”) comment is generic and not supported by any reason or discussion.*

*With reference to the higher complexity (“**more complex**”) comment, we never claimed that the proposed approach can, immediately and for all possible applications, replace the conventional ICs (integrated circuits) technology, which, after decades of enormous research efforts, nowadays offers outstanding speed, reliability (e.g. systems containing millions of devices running flawlessly for many years at billions of cycles per second), compactness, light weight, power efficiency...*

On the contrary, we have explicitly mentioned, as a remarkable advantage, that the proposed approach is compatible with CMOS and MEMS processes.

We have not claimed a generic superiority, but have given clear and specific advantages (time variant implementation of gates instead of unchangeable gates) which we have experimentally demonstrated (e.g. Fig. 5) and may not be achieved with existing technologies.

Moreover, the higher complexity (“**more complex**”) comment seem quite vague as any innovative technology may be somehow be considered as “**more complex**” than established technologies.

Finally, the Reviewer #3 generically and hypothetically states that the proposed approach “**might be... even impossible**”, but did not provide any concrete reason, objection or discussion to support this opinion. We confirm that the very-large-scale integration of devices drawn by tunneling-triboelectrification is possible for the following key reasons (see the manuscript for more details):

- a) Fig. 5 experimentally demonstrates the very-large-scale integration (with deep sub-micron resolution) of ghost floating gates by means of a conventional AFM tip
- b) as stated in the manuscript, “AFMs can be fully integrated on a single chip²⁵⁻²⁷”
- c) as stated in the manuscript, “we envision single chips comprising the AFM and regions of 2D materials whose properties can be controlled on demand by tunneling triboelectrification. This approach would be similar to the memory refresh process (i.e. periodically reading and immediately rewriting the same information on capacitive memories in order to counteract the degradation of information due to leakage currents) routinely used in dynamic random-access memories (DRAM).”

References

- [1] Lemme M. C. *et al.*, A Graphene Field-Effect Device. *IEEE Electron Device Letters* **28**, 282–284 (2007).
- [2] Schedin F. *et al.*, Detection of individual gas molecules adsorbed on graphene, *Nature Materials* **6**, 652–655 (2007).
- [3] Meric I. *et al.* Current saturation in zero-bandgap, top-gated graphene field-effect transistors, *Nature Nanotechnology* **3** (2008).
- [4] Grassi *et al.*, Boosting the voltage gain of graphene FETs through a differential amplifier scheme with positive feedback, *Solid State Electronics* **100**, 54–60 (2014).
- [5] Han S. *et al.*, Microscale loop-mediated isothermal amplification of viral DNA with real-time monitoring on solution-gated graphene FET microchip, *Biosensors and Bioelectronics* **93**, 220-225 (2017).
- [6] Smith A. D. *et al.*, Resistive graphene humidity sensors with rapid and direct electrical readout, *Nanoscale* **7**, 19099–19109 (2015).
- [7] Chun S. *et al.*, A graphene force sensor with pressure-amplifying structure, *Carbon* **78**, 601–608 (2014).
- [8] Khan U. *et al.*, Self-powered transparent flexible graphene microheaters, *Nano Energy* **17**, 356–365 (2015).
- [9] Bae S. H. *et al.*, Graphene-based transparent strain sensor, *Carbon* **51**, 236-242 (2013).
- [10] Chun S. *et al.*, All-graphene strain sensor on soft substrate, *Carbon* **116**, 753-759 (2017).